

**Applying IT Communication Technology in Public Awareness**
**and Education for Reducing Hazard Casualty in South East**
**Asia Developing Countries**
S. P. Koay[1], L. T. Tay[1], H. Y. Chan[1], S. Jamaludin[2], H. Fukuoka[3], H. Hazarika[4], N. Sakai[5],
H. Lateh[1]
[1]Universiti Sains Malaysia, Penang 11800, Malaysia
[2]Slope Engineering Branch, Public Works Department, Kuala Lumpur 50582, Malaysia
[3]Research Institute for Natural Hazards and Disaster Recovery, Niigata University, Niigata 950-2181, Japan
[4]Faculty of Engineering, Kyushu University, Fukuoka 819-0395, Japan
[5]National Research Institute for Earth Science and Disaster Prevention, Tsukuba 305-0006, Japan
*Correspondence to*: S. P. Koay (spkoay@cs.usm.my)
**Abstract.** In developing countries, especially most of South East Asia countries, every year landslides,
mudslides and flood occur during monsoon rainy season and cause casualties not only in rural areas, but also in
urban areas. Public awareness and education activities are important to reduce the casualties of such natural
disasters. Nowadays, IT communication technology plays an important role in disseminating information and
education. Lately, applying IT communication technology for public awareness on natural hazard becomes a
trend among government authorities in these countries.
We begin our study in Malaysia on how to reduce landslides casualty for future natural hazard management in
these developing countries. For a better understanding on the occurrence of landslides, besides the mechnism on
how landslides occur, collecting historical data with location coordinates were carried out and stored in database
periodically. Public can browse these historical data via internet and know which areas are more prone to
landslides occurrence.
At the same time, IT communication technology was used to disseminate alert information after receiving
rainfall intensity data from the monitering sites. It was also more efficient to conduct the public and school
children awareness education by using such technology, as the simulation of rainfall induced slope failure
mechanism and educational video on symptoms of landslides before they occur may give a clearer picture and
better understanding to the public and school children.



**1 Introduction**

Every year, the Northeast Monsoon brings heavy rain to the East Coast of Peninsular Malaysia from November to March. In 2014, heavy rain (recorded 890mm rainfall from 19th. December, 2014 to 27th. December, 2014) caused flood and landslides along the East Coast of Peninsular Malaysia. The East-west Highway (Federal Route 4) in Malaysia was cut off due to slope failures on 23rd. December, 2014 as shown in Fig. 1. It caused casualties and billion of Malaysian Ringgit economical losses. Moreover, according to the news, in 2015, besides the 3 large scale landslides that happened in Malaysia, landslides were also reported in Philipines(2), Indonesia(3), Thailand(1), Myanmar(1) and Vietnam(2).

In July 2015, the City Hall of Kuala Lumpur pointed out that there are more than 600 slopes which are classified as high risk slopes, after conducting sites survey in Kuala Lumpur, with the Public Works Institutes of Malaysia. Therefore, to reduce the occurrence of casualty, the public awareness and education on natural hazard should be given priority.

In Motoya, E. et al. (2015) study, they only carried out crowdsourcing study for community mapping towards public awareness and preparedness on the disaster occurrence, mainly slope failure. Ahmad, J. and Lateh, H. (2015) discussed on how important the role of teachers in educating students on the risk of landslides disaster in rural and urban areas.

Here, besides introducing how IT technology was used to disseminate early warning information, landslides education workshops in primary schools were conducted to provide students with a better understanding on natural hazard, for reducing natural disaster casualty.

**2 IT technology in collecting monitoring data and disseminating early warning information**

Most of slope failures in South East Asia occurred after heavy rain. Rainfall was the main factor that induced landslides. Slope failures were also caused by unplanned development of hilly areas which made way for housing projects due to the ever growing population in the cities in these fast growing countries. Therefore, a good monitoring system such as the tipping bucket rain gauge installation is needed to monitor the high risk slopes. In our study, we installed 2 main monitoring systems along the East-west Highway, Malaysia where the locations are N05 ° 32.918' E101 ° 20.749' and N05° 36.042' E101° 35.546'. The collected number of tipping times from the sites are sent to the monitoring server and converted to rainfall intensity value by multiplying 0.5mm.

In our study on landslide prediction, if the curve in the accumulated rainfall vs rainfall intensity graph crosses over the cautious and critical lines, Koay, S. P. et al. (2013), as shown in Fig. 2, the alert message will be sent to the pre-registered and authorized users for early warning dissemination purpose via email and SMS to mobile phone as in Fig. 3.



The cautious and critical lines were generated by the simulator, Koay, S. P. et al.(2008), with the condition of the slope angle > 33 °, and the soil properties: cohesion $Ca$ = 25g/cm², effective porosity / valid porosity λ = 0.4, saturated hydraulic conductivity K = 0.02 m/h, internal friction angle $\Phi$ = 35 ° and soil unit weight $\gamma_s$ = 1.36 g/cm³. Unlike developed countries, at present, there is not enough rainfall-induced historical landslides data to create the cautious and critical lines to predict slope failure, in Malaysia. These lines will be continuously calibrated with the real rainfall data that triggered the landslides.

**3 Natural hazard historical information and education for public awareness**

Well-designed early warning system is not enough to reduce casualties. Public awareness on natural hazard is also very important to avoid casualties. More information on natural hazard historical occurrence venue (Fig. 4), knowledge on the mechanism and symptoms of landslides occurrence, and hazard map for preparedness of evacuation to shelter (Fig. 5) if natural hazard occurs, via internet, are needed for gaining the public awareness. The well-informed and well-prepared public will not panic when confronting the natural hazards. During 2011 Tohoku Earthquake and Tsunami Disaster, Iwate Prefecture has less number of mortality compared to Miyagi Prefecture as Iwate Prefecture local government practised more public awareness education and evacuation training in schools, according to past disaster experiences, Hayashi H. (2015). Therefore, education and learning on how to obtain the natural hazard information should be practiced since childhood. Promoting natural hazard education, through games, for school children learning in primary school curriculum is recommended, especially where the schools are located in the hilly area. Besides showing the slides or videos on natural disaster, hands-on education should be conducted in schools as Koay, S. P. et al. (2015).

Mixed terminology can confuse the public, while too many technical words, jargons, or scientific language is unhelpful for those who do not know much about geological hazards, Gill, J. et al.(2015), in the primary school natural hazard education. Using simple terms may give school children a better understanding, as the main purpose of the education is to teach the students.

A natural hazard education workshop was conducted on 7[th] April, 2015 in a school, SK RPS Banun, in rural area. Most of the students are aborigines Jahai, Negrito tribe. Later on 12[th] October, 2015 and 2[nd] November, 2015, we conducted 2 more workshops in schools in urban area. One is SJKC Perempuan China, where the students are good in Chinese language and the other is SJK Minden Height, where the students understand Malay language well. Before starting the explanation on landslides, questionnaires on landslides knowledge and weather station were given to the students to answer. Less than 5% students managed to give the right    answer in the rural area while more than 78% students from the urban area gave the correct answer. None of the students from SK RPS Banun could understand the high risk of landslides road signboard with only the simple landslides symbol. However, after showing them the photo of the signboard with slope failure at the back, all the students could answer. It is clear that the signboard photos with the slope failure as background, provides a better understanding.



From our questionnaire survey, most of the school students in the urban area have knowledge on how and why
natural hazard happens before conducting education workshop compared to the rural area students. However,
after attending 30 minutes of simple slides show on what would happen after heavy rain in the hilly area,
landslides symptom by comparing the colour of spring water from the soil in the slope in two videos Koay, S. P.
et al. (2015) as Fig. 6, slope failure mechanism, and hands-on weather station experiments as Fig. 7, the school
students in the rural area gave right answers to the questions and understood the functions of a weather station
well in the post education workshop questionnaire. Moreover, after taking the natural hazard class, all of the
students from rural and urban areas, were more alert of the natural hazards in their surroundings. This natural
hazard education workshop managed to attract the students' attention. They are more prepared to face the natural
hazards now and also looking forward to learning more on natural hazards in the future.
We also concluded that it is important to use the students' mother tongue or first language to conduct the
workshop, especially in primary schools, for providing a better picture to the students after conducting the
workshop in SJKC Perempuan China by using the Malay language, where most of the students could not
understand well in the Malay language even though it is national official language in Malaysia, as shown Table
1. In most of the South East Asia countries, there are many races who speak their own languages when at home.
We should be aware of the language while conducting education workshop in primary schools in these
developing countries.
**4 Conclusions**
IT communication technology plays a major role in predicting the occurrence of landslides and disseminating
information to the public at the right time and the exact venue. It is more effective to conduct school children
education on natural hazard by using IT technology.  Choosing the right mother tongue or first language and
using simple terms to explain to the students are very important for better understanding when conducting
natural hazard education in primary schools in developing countries where there are many races. Education on
natural hazard should be conducted in schools, especially in lakeside, riverside and hilly areas. Learning natural
hazard by playing games and hands-on experiment may attract students' attention in the primary schools.
**Acknowledgements**
Financial support provided by Grants: The Project For Research and Development for Reducing Geo-Hazard
Damage in Malaysia Caused by Landslide and Flood, JICA, MOE and Universiti Sains Malaysia
(203/PJJAUH/6711279) are gratefully acknowledged. We would also like to thank Mr. Ewe Hoe Tan, Dato' Dr.
Mohd Razha Adb. Rashid, Dr. Tomofumi Koyama, Dr. Satoshi Murakami, Mr. Mitsuru Yabe and other
colleagues for giving us their generous collaboration in this study.



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

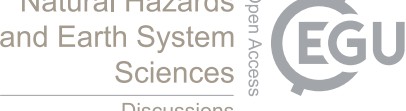



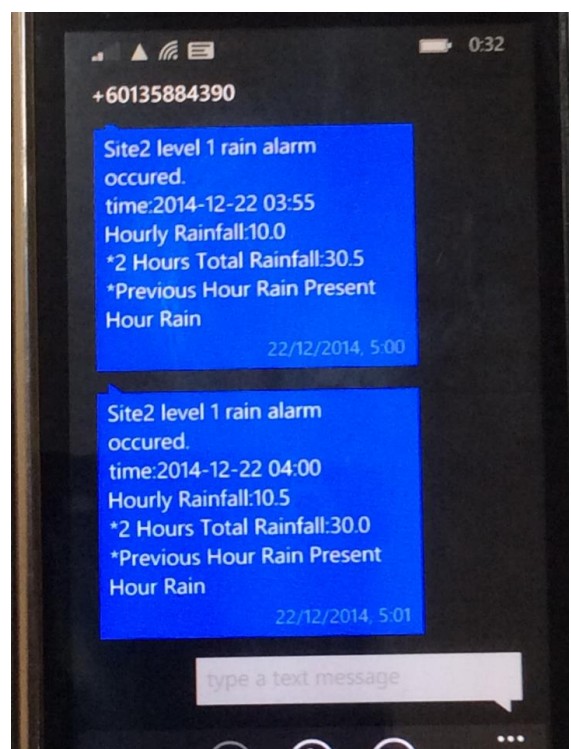

2      Figure 3. Alert message was sent to the registered user via SMS.





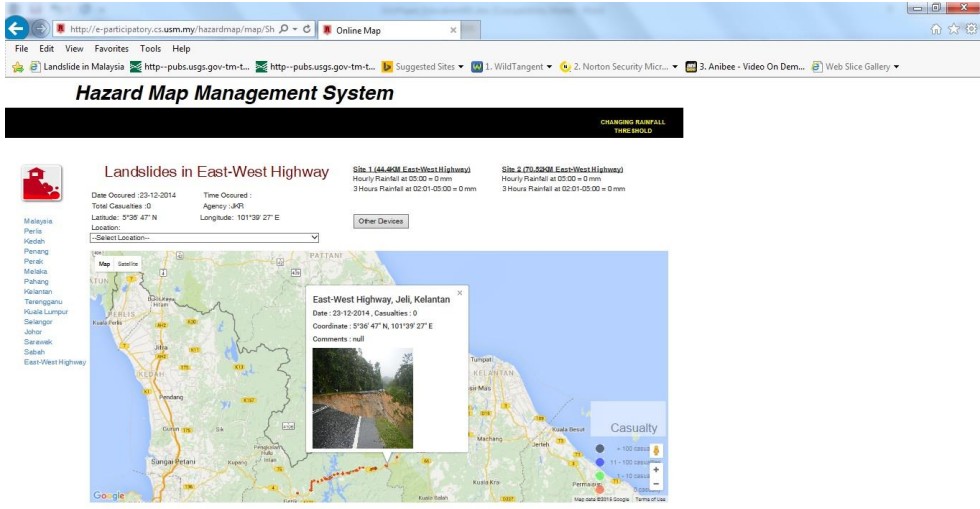

2    Figure 4. Historical data providing natural hazard information on a hazard map.





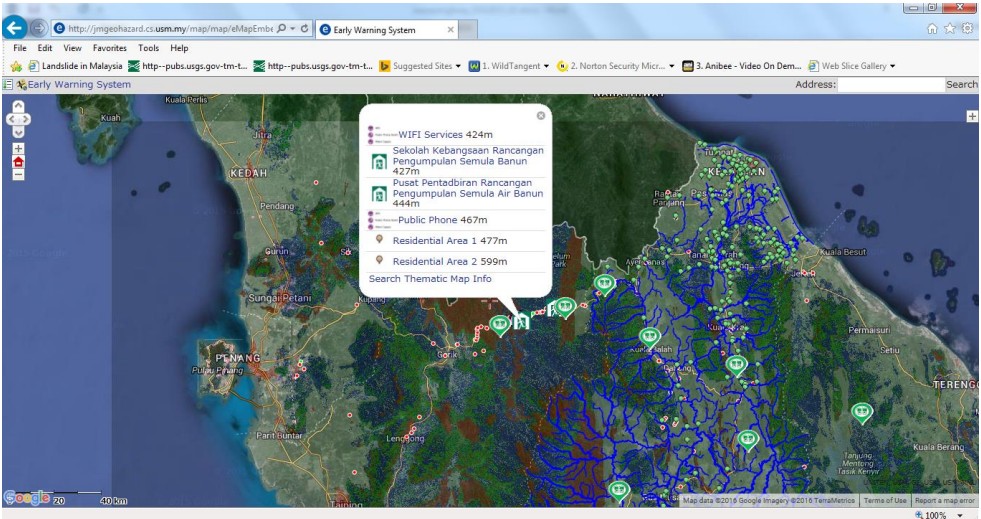

2    Figure 5.  Hazard map for evacuation to shelter along East-west Highway.



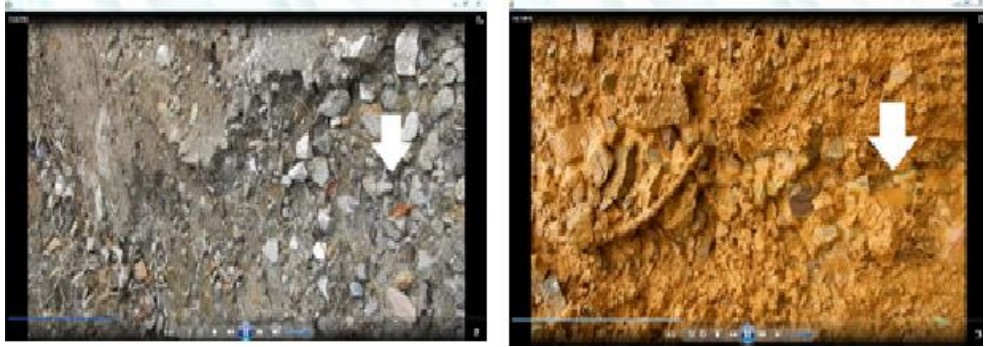

Figure 6. Video showing a sound slope with pure spring water and a slope at high risk with impure spring water





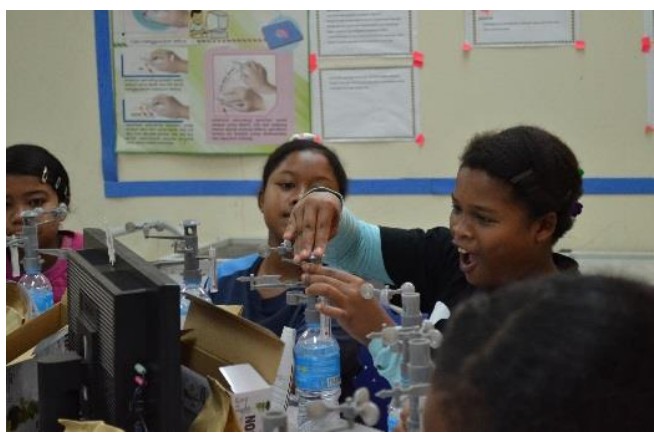

2    Figure 7. Hands-on education on weather station.



| Number of correct answers to questions on the function of a Weather Station | | | |
|---|---|---|---|
| | SK RPS Banun(school in rural area)<br>(Malay Language) | SJKC Perempuan China(school in urban area)<br>(Chinese Language) | SJK Minden Height(school in urban area)<br>(Malay Language) |
| Before Explanation(in Malay Language) | 1 correct (out of 50 students) | 39 correct (out of 50 students) | 47 correct (out of 50 students) |
| After Explanation(in Malay Language) | 43 correct (out of 50 students) | 42 correct (out of 50 students) | 48 correct (out of 50 students) |

3        Table 1. Result of answering same question before and after education.