# Peer review of "Applying IT Communication Technology in Public Awareness"

_Natural Hazards and Earth System Sciences, 2016_

## Referee Comment (RC1) · Anonymous Referee #1 · 19 Jan 2016

Review of Applying IT Communication Technology in Public Awareness and Education for Reducing Hazard Casualty in South East Asia Developing Countries.

The subject area and intent of this paper are both very interesting, but this paper is not at a level that can be considered for an international journal. My major comments are:

(a) Understanding of the literature. The paper's main aim is to consider IT Communications in a context of SE Asia hazards. This is a very good goal, but there is a large body of literature which needs to be acknowledged that examines IT Communications in a broader natural hazards context, and then where this study sits within it. Of those items

cited (eight references) the vast majority are abstracts from conferences—this is not suitable for an international publication. The authors need to significantly understand the literature, and where to place their fundamental observations and questions within this literature. (b) Research methodology. There is not a clear research methodology here. Is it the questionnaire (and if so, what were the questions)? How many students were there? Was there a methodology if how these particular classes were chosen?

This manuscript reads much more like an extended abstract for a conference proceedings, which certainly has some interesting goals, but little substance of background understanding of the issues involved or presentation of the research methodology. If the authors can substantially revise their paper, it might be submitted again, but should not be accepted (even for major revision) as is, as it is not yet ready to be reviewed.

---

## Referee Comment (RC2) · Anonymous Referee #2 · 27 Jan 2016

The paper addresses an interesting scientific question related to Natural Hazards – IT communications and how they can be used for public awareness of hazards. This is an interesting topics from several points of view: hazards, communications, engagement and computer science application.

The paper alludes to some interesting new tools and techniques, including the image of the alert message sent to a registered user, but there is no formal description of this or user evaluation study and it is not clear whether this alert system was designed by the authors themselves. The videos and other materials used in the school workshops are

interesting. The paper seems very short, with some interesting ideas but without these being situated in the literature, methods or detailed results and analysis. There needs to be a larger literature review and methodology. The literature review relies mainly on conference presentations and published abstracts, and does not draw on the wider peer-review literature. Regarding the methods used - a questionnaire is mentioned but there is no description of all the questions. The landslide education workshops are mentioned and activities outlined but there needs to be more definite information such as how many students were involved and how old were the students? There is not enough information given here for reproducibility.

The paper reads like an outline or extended abstract, and needs to be elaborated upon substantially, drawing upon a wider range of literature. There are some places where the language is not as clear as it could be in places. Careful consideration and explanation needs to be given to explaining what the authors mean by IT and what parts of IT they are using and for what purpose. The title could be reconsidered as it implies work across several countries in South East Asia and in the general public as opposed to school children.

The figures are useful in giving a flavour of the schools workshop activities but are rather small meaning that it is hard to see the content of the predication tools and maps.

Overall the paper needs to be longer, containing more relevant literature and a more in-depth methodology, results and analysis. In it's current state it is hard to properly evaluate the scientific quality and significance of the work and whether it is suitable for publication which requires more than major revisions.

---

## Author Comment (AC1) · 1 Feb 2016

We would like to reply to the comments of referee #1 as below:

A) This is a very good goal, but there is a large body of literature which needs to be acknowledged that examines IT Communications in a broader natural hazards context, and then where this study sits within it. Of those items cited (eight references) the vast majority are abstracts from conferences. This is not suitable for an international publication. The authors need to significantly understand the literature, and where to place their fundamental observations and questions within this literature.

There are several IT Communications literatures and reports that we did refer, some are listed in the references of our previous publications. Those literatures will be added as references in the revised version of this paper in addition to our previous publications. There are also a few abstracts in our references and I attended the presentation in the conferences of these abstracts. The research information obtained during their presentation are referred in our study. We would also like to add in the sentences (references) for example:

1) In 2007, Vyas foresaw by using IT Technology such as GIS, Remote Sensing, Internet and Warning System will reduce casualty in natural and man-made disaster. (Reference: Vyas, T. et al : Information Technology for Disaster Management, Proceedings of National Conference INDIACom-2007, Computing for Nation Development, 2007.)

2) Social media such as Facebook, YouTube, Twitter and so on, were used to give warning to reduce the risk of disaster in Dufty's study. (Reference: Dufty, N. :The use of social media in countrywide disaster risk reduction public awareness strategies, Australian Journal of Emergency Management, Volume 30 Issue 1 Articles: 26, 2015.)

3) Lai reported that the number of smart mobile phone users is increasing in Indonesia, Myanmar, Philippines and Vietnam. The usage of such mobile device to disseminate warning message becomes quite common. However, still most of these countries citizens still refer to radio/TV news stations as a reliable source. (Reference: Lai, C. et al: State of the use of Mobile Technologies for Disaster Preparedness in South East Asia, Report by Nanyang Technological University, Singapore, 2015.)

For the acknowledged references 1) "Japanese Experience with Long-term Recovery from the 2011 Tohoku Earthquake and Tsunami Disaster" by Hayashi, H., I attended his presentation. In his presentation contents, data which showed that Iwate Prefecture(4,673 death), Japan has less casualty than Miyagi Prefecture(9,541 death), Japan can be obtained from http://www.bousai.go.jp/2011daishinsai/pdf/torimatome20150909.pdf    (in    Japanese

Language) page 37/154 (National Research Institute for Earth Science and Disaster Prevention, Japan report), and public awareness education and evacuation training were carried out continuously in Iwate Prefecture is reported in page 36/258 in http://www2.pref.iwate.jp/~bousai/kirokushi/allpage.pdf (in Japanese Language)(Iwate Prefecture Government report in Japan).

2) "Natural Hazards Education, Communications and Science-Policy-Practice Interface, SPM1.43", it is a note which was prepared by Gill, J., Malamud, B. D., Taylor, F., Mohadjer, S, and Charrière , M., after EGU 2015 Workshop and can be obtained via www.groupspaces.com/SocialGeoscience/ .

3) The contents of "Study of rain induced landslides prediction and casualty prevention in Malaysia", ISM Symposium on Environmental Statistics 2015, Tokyo, 2015 are mostly based on "The Prediction of Water Table Flow in Slope for Early Warning System in Malaysia" (6th International Geotechnical Symposium on Disaster Mitigation in Special Geoenvironmental Conditions (6IGS Chennai 2015)) pp 491 – 494 and "Information Technology for Disaster Management", (National Conference INDIACom-2007, Computing for Nation Development, 2007).

4) The references of "Study of Disseminating Landslide Early Warning Information in Malaysia", EGU General Assembly 2015, Vienna, 2015 are 1. Hiramatsu Shinya, Mizuyama Takahisa, Ogawa Shigeru, Ishikawa Yoshiharu, (1992) Influence of Rainfall Time Distribution on Shallow Landslides. Japan Society of Erosion Control Engineering Vol.44 No.5, Ser. No.178 2. Komamura Fujiya, (1988) Estimation of Critical Volume of Rain to Surface Failure Occurrence. Journal of Japan Landslide Society 25-1 3. Koay Swee Peng, Lateh Habibah, Sakai Naoki, Morohoshi Toshikazu and Fukuzono Teruki, (2008), The Preliminary Study on Landslide Prediction Model in Malaysia. The First World Landslide Forum, ICL 2008, Tokyo, pp. 493-498 4. Koay Swee Peng, Lateh Habibah, Murakami Satoshi, Koyama Tomofumi, Sakai Naoki and Jamaludin Suhaimi (2014), Slope Monitoring and Landslide Disaster Mitigation in Kyoji Sassa et. al. (editors), Landslide Science For A Safer Geoenvironment, Volume 4, The International Programme on Lndslides (IPL), Publisher: Springer. ISBN 978-3-319-04998-5 5. Nunokawa Osamu, Sugiyama Tomoyasu, Ota Naoyuki, Hata Akihito, Hori Michihiro, Kamemura Katsumi and Okada Katsuya, (2010) A Method To Calculate Expected Frequency of Rainfall-Induced Slope Failure Considering Train Operation Control. Journal of Japan Society of Civil Engineers Ser. C Vol. 66 No. 1, pp. 78 – 88

5) Most of the contents in "The Study on Landslide Disaster Mitigation and Management Using Numerical Analysis in Malaysia, Japan Geoscience Union Meeting 2013, Makuhari Messe, Chiba, 2013" are from Landslide Prediction Using Numerical Analysis, Caspian Journal of Applied Sciences Research, 2(AICCE'12 & GIZ' 12), 2013, pp. 336-342, http://www.cjasr.com ISSN: 2251-9114, 2012 CJASR

B) Research methodology. There is not a clear research methodology here. Is it the questionnaire (and if so, what were the questions)? How many students were there? Was there a methodology if how these particular classes were chosen?

1) We appreciate the comments from the referee. We would provide the detailed information of our research methodology in the revised version of this paper. Below is the answer for the referee: Please refer to attached files for the questionnaires (original and translated in English Language) and, workshop contents and questions during the workshop (original and translated in English). We conducted 3 education workshops. There were 50 students in SK RPS Banun(1st workshop), 220 students in SRKC Perempuan Cina(2nd workshop), 150 students in SJK Minden Height(3rd workshop). We randomly picked 50 students answers from SRKC Perempuan Cina and SJK Minden Height to make it the same number of students as in SK RPS Banun. We requested the headmaster and headmistress to select 11 years old to 12 years old students. They assigned Primary Year 6 Students to attend our workshop in the schools.

We would like to apologise if our above reply to the referee is not suitable or still insufficient.

Please also note the supplement to this comment:
http://www.nat-hazards-earth-syst-sci-discuss.net/nhess-2016-15/nhess-2016-15-AC1-supplement.pdf

[Figure]

**Fig. 1.**

[Figure]

**Fig. 2.**

[Figure]

**Fig. 3.**

**Fig. 4.**

**QUESTION AFTER LANDSLIDES EDUCATION**

Location          : (school name)
Instruction       : Please tick (√) in every box

**A.   Demographics**

1. Age  :
2. Sex  :          ☐ Male          ☐ Female

**B.   Knowledge on Landslides**

3. Did landslides happen near to your house?   ☐ Yes          ☐ No
4. Landslides happened mostly because of
   ☐ Heavy rain
   ☐ Earth quake
5. Which will trigger landslides
   ☐ Strong wind
   ☐ Heavy rain
   ☐ Lightning
6. Among the below natural disaster, which often in your place?
   ☐ Flood
   ☐ Landslides
   ☐ Thunder
   ☐ Typhoon

7. Is it important to know about mechanism of landslide?        ☐ Yes   ☐ No
8. Should I know often getting more knowledge on landslides     ☐ Yes   ☐ No
9. Landslides cause losses of money and casualties              ☐ Yes   ☐ No

**C.   Readiness**

10. I became self-prepared to face landslides in future.                       ☐ Yes   ☐ No
11. Knowledge on the happening of landslides is very important as it will save my life   ☐ Yes   ☐ No

**QUESTION ON LANDSLIDES STUDY**

12. Fill in the blank with the number

1.  Thermometer is for measuring air temperature
2.  Rain gauge is for measuring rainfall intensity
3.  Anemoscope is for pointing the wind direction
4.  Compass is for getting the direction
5.  Anemometer is for measuring wind speed

13. If you have a chance to attend such education, will you attend?
    1. Yes   ☐
    2. No    ☐

---

## Author Comment (AC2) · 16 Feb 2016

(1) The paper alludes to some interesting new tools and techniques, including the image of the alert message sent to a registered user, but there is no formal description of this or user evaluation study and it is not clear whether this alert system was designed by the authors themselves.

We developed 2 types of alert systems with different alert threshold value settings. We then monitored which alert system is more sensitive to the occurrence of landslides and gives a better landslides prediction.
[Figure]

a) Type 1: Alert Level 1(cautious): If rainfall intensity is more than 10mm/hour and 2 hours accumulated rainfall is more than 30 mm

Alert Level 2(preparing for evacuation): If rainfall intensity is more than 30mm/hour or 2 hours accumulated rainfall is more than 50mm

Alert Level 3(evacuation): If rainfall intensity is more than 50mm/hour or 2 hours accumulated rainfall is more than 80mm

If there is no rain for 12 continuous hours, withdraw the warning and reset the accumulative rainfall.

The alert system will send SMS warning message to the pre-registered officer in charge. These threshold values are set after studying the previous rainfall intensity during landslides occurrences, with the advice from the Public Works Department, Malaysia.

b) Type 2: We use the graph accumulated rainfall (axis x) vs. rainfall intensity (axis y) method to predict the risk of the slope failure. If the accumulated rainfall line crosses over the cautious line and critical line, Koay, S. P. et al. (2013), the message will be disseminated to the pre-registered officer in charge for decision making. We set cautious line of axis x value 90% less than the critical line axis x value after getting advice from the officer from the Public Works Department, Malaysia. Moreover, the line in the graph will be reset to the origin if there is no rain for 24 continuous hours or the rainfall intensity is less than 1mm/h during 24 hours.

Reference: Swee Peng Koay, Habibah Lateh, Satoshi Murakami, Tomofumi Koyama, Naoki Sakai and Suhaimi Jamaludin, 2014, Slope Monitoring and Landslide Disaster Mitigation in Malaysia, Kyoji Sassa et. al. (editors), Landslide Science For A Safer Geoenvironment, Volume 4, The International Programme on Landslides (IPL), pp 318 - 324

(2) There needs to be a larger literature review and methodology. The literature review

relies mainly on conference presentations and published abstracts, and does not draw on the wider peer-review literature.

The below is the reply to referee #1. Please allow us to reuse them.

There are several IT Communications literatures and reports that we referred to, some are listed in the references of our previous publications. Those literatures will be added as references in the revised version of this paper in addition to our previous publications. There are also a few abstracts in our references and I attended the presentation in the conferences of these abstracts. The research information obtained during their presentation are referred in our study. We would also like to add in the sentences (references), for example:

1) In 2007, Vyas foresaw by using IT Technology such as GIS, Remote Sensing, Internet and Warning System will reduce casualty in natural and man-made disaster. (Reference: Vyas, T. et al : Information Technology for Disaster Management, Proceedings of National Conference INDIACom-2007, Computing for Nation Development, 2007.)

2) Social media such as Facebook, YouTube, Twitter and so on, were used to give warning to reduce the risk of disaster in Dufty's study. (Reference: Dufty, N. :The use of social media in countrywide disaster risk reduction public awareness strategies, Australian Journal of Emergency Management, Volume 30 Issue 1 Articles: 26, 2015.)

3) Lai reported that the number of smart mobile phone users is increasing in Indonesia, Myanmar, Philippines and Vietnam. The usage of such mobile device to disseminate warning message becomes quite common. However, still most of these countries citizens still refer to radio/TV news stations as a reliable source. (Reference: Lai, C. et al: State of the use of Mobile Technologies for Disaster Preparedness in South East Asia, Report by Nanyang Technological University, Singapore, 2015.)

For the acknowledged references

1) "Japanese Experience with Long-term Recovery from the 2011 Tohoku Earthquake

and Tsunami Disaster" by Hayashi, H., I attended his presentation. In his presentation contents, data which showed that Iwate Prefecture(4,673 death), Japan has less casualty than Miyagi Prefecture(9,541 death), Japan can be obtained from

http://www.bousai.go.jp/2011daishinsai/pdf/torimatome20150909.pdf (in Japanese Language) page 37/154 (National Research Institute for Earth Science and Disaster Prevention, Japan report), and public awareness education and evacuation training were carried out continuously in Iwate Prefecture is reported in page 36/258 in

http://www2.pref.iwate.jp/~bousai/kirokushi/allpage.pdf (in Japanese Language)(Iwate Prefecture Government report in Japan).

2) "Natural Hazards Education, Communications and Science-Policy-Practice Interface, SPM1.43", it is a note which was prepared by Gill, J., Malamud, B. D., Taylor, F., Mohadjer, S, and Charrière , M., after EGU 2015 Workshop and can be obtained via www.groupspaces.com/SocialGeoscience/ .

3) The contents of "Study of rain induced landslides prediction and casualty prevention in Malaysia", ISM Symposium on Environmental Statistics 2015, Tokyo, 2015 are mostly based on "The Prediction of Water Table Flow in Slope for Early Warning System in Malaysia" (6th International Geotechnical Symposium on Disaster Mitigation in Special Geoenvironmental Conditions (6IGS Chennai 2015)) pp 491 – 494 and "Information Technology for Disaster Management", (National Conference INDIACom-2007, Computing for Nation Development, 2007).

4) The references of "Study of Disseminating Landslide Early Warning Information in Malaysia", EGU General Assembly 2015, Vienna, 2015 are

1. Hiramatsu Shinya, Mizuyama Takahisa, Ogawa Shigeru, Ishikawa Yoshiharu, (1992) Influence of Rainfall Time Distribution on Shallow Landslides. Japan Society of Erosion Control Engineering Vol.44 No.5, Ser. No.178

2. Komamura Fujiya, (1988) Estimation of Critical Volume of Rain to Surface Failure

Occurrence. Journal of Japan Landslide Society 25-1

3. Koay Swee Peng, Lateh Habibah, Sakai Naoki, Morohoshi Toshikazu and Fukuzono Teruki, (2008), The Preliminary Study on Landslide Prediction Model in Malaysia. The First World Landslide Forum, ICL 2008, Tokyo, pp. 493-498

4. Koay Swee Peng, Lateh Habibah, Murakami Satoshi, Koyama Tomofumi, Sakai Naoki and Jamaludin Suhaimi (2014), Slope Monitoring and Landslide Disaster Mitigation in Kyoji Sassa et. al. (editors), Landslide Science For A Safer Geoenvironment, Volume 4, The International Programme on Lndslides (IPL), Publisher: Springer. ISBN 978-3-319-04998-5

5. Nunokawa Osamu, Sugiyama Tomoyasu, Ota Naoyuki, Hata Akihito, Hori Michihiro, Kamemura Katsumi and Okada Katsuya, (2010) A Method To Calculate Expected Frequency of Rainfall-Induced Slope Failure Considering Train Operation Control. Journal of Japan Society of Civil Engineers Ser. C Vol 66 No. 1, pp. 78 – 88

5) Most of the contents in "The Study on Landslide Disaster Mitigation and Management Using Numerical Analysis in Malaysia, Japan Geoscience Union Meeting 2013, Makuhari Messe, Chiba, 2013" are from Landslide Prediction Using Numerical Analysis, Caspian Journal of Applied Sciences Research, 2(AICCE'12 & GIZ' 12), 2013, pp. 336-342, http://www.cjasr.com ISSN: 2251-9114, 2012 CJASR

(3) Regarding the methods used - a questionnaire is mentioned but there is no description of all the questions. The landslide education workshops are mentioned and activities outlined but there needs to be more definite information such as how many students were involved and how old were the students? There is not enough information given here for reproducibility.

Please refer to attached files for the questionnaires (original and translated in English Language, Fig. 1 – Fig. 4) as well as workshop contents and questions during the workshop (original and translated in English, refer to Supplement). We conducted

3 education workshops. There were 50 students in SK RPS Banun(1st workshop), 220 students in SRKC Perempuan Cina(2nd workshop), 150 students in SJK Minden Height(3rd workshop). We randomly picked 50 students' answers from SRKC Perempuan Cina and SJK Minden Height to make it the same number of students as in SK RPS Banun. We requested the headmaster and headmistress to select 11 years old to 12 years old students. They assigned Primary Year 6 Students to attend our workshop in the schools.

(4) Careful consideration and explanation needs to be given to explaining what the authors mean by IT and what parts of IT they are using and for what purpose. The title could be reconsidered as it implies work across several countries in South East Asia and in the general public as opposed to school children.

Thank you for your advice. Actually in our proceeding with the title "Application of Signal Processing Technology in Monitoring High Risk Slope for Landslides Prediction in Malaysia " in "2015 RISP International Workshop on Nonlinear Circuits, Communications and Signal Processing (NCSP'15)", we discussed on how electronics devices detect the movement of slopes by measuring the electricity current and voltage. The measured readings were collected in the data logger before sending to the server by modem via internet. Furthermore, the collected data, from monitoring system, were processed and analysed by an early warning system which was developed by us, in the workstation(Fig. 5). If the curve in accumulated rainfall (axis x) vs. rainfall intensity (axis y)(Fig. 6) crosses the cautious line and critical line, the alert message will be disseminated to the person in charge via SMS and email. The light turns to yellow in colour and soft buzzer sound turns on if the curve crosses over the cautious line, and the light turns to red in colour and hard buzzer sound turns on if the curve crosses over the critical line.

Besides applying IT technology on data transmission and information dissemination, we also designed a workstation simulator to analyse how the increment of water table caused by rainfall intensity and the slope failure, Koay, S. P. et al.(2008). User

can input the necessary data, for example rainfall intensity, slope angle, and the soil properties: cohesion, effective porosity, valid porosity, saturated hydraulic conductivity, internal friction angle and unit weight, to run the simulation for better understanding of relations among rainfall intensity, water table and slope stability with the same soil properties. Moreover, the user can also change the slope angle and soil properties by own scenario for analysis purposes.

Please refer to Fig. 7 for the image of the simulator. Furthermore, regarding to applying which model to develop the simulator, please refer to Landslide Prediction Using Numerical Analysis, Caspian Journal of Applied Sciences Research, 2(AICCE'12 & GIZ' 12), 2013, pp. 336-342, http://www.cjasr.com ISSN: 2251-9114, 2012 CJASR. We try not to show equations here, for the benefit of readers who do NOT have civil engineering background. It is easier for them to read the paper and they can refer to the reference paper if necessary.

(5) The figures are useful in giving a flavour of the schools workshop activities but are rather small meaning that it is hard to see the content of the predication tools and maps

We apologize for NOT considering user interface while submitting the manuscript. Please refer to Fig. 8 for the hazard map management. Moreover, please access to the below URL

http://e-participatory.cs.usm.my/hazardmap/map/Show.aspx?country=Highway

for clear picture of hazard map management system, which provides landslides historical information to users.

(6) The title could be reconsidered as it implies work across several countries in South East Asia and in the general public as opposed to school children.

Outreach workshops conducted in school classrooms to teach students and teachers about the aspect of science (natural hazard) in an engaging manner stimulate the interest and indirectly awareness of learners on natural disasters (Illingworth et al.

(2015)).

Most of the primary schools are designated as shelters from disaster in local community, especially in Japan (Nagamatsu et al. (2009)). Therefore, we started our hazard education workshop in primary schools, and it is easier to explain to students about the disaster shelters in case of disaster occurrence.

Moreover, most of the parents are busy in their work in South East Asia countries, especially in Malaysia. They may not have the time to attend the natural hazard education workshop. Furthermore, in rural areas, some parents are not well-educated. Hence, we started natural hazard education workshop for students in primary schools hoping that the school children will discuss the topics with their family while having conversation among their family members, for public awareness on natural disasters.

In addition, while carrying on the hazard education workshop, we tried to avoid using scientific lexicons; instead we showed more photos and pictures for better understanding, as most of scientific lexicons are overly technical and the general public may not get the clear picture (Stewart et al. (2013)).

References: 1. Samuel M. Illingworth and Heidi A. Roop, (2015) Developing Key Skills as a Science Communicator: Case Studies of Two Scientist-Led Outreach Programmes. Geosciences 2015, 5, pp. 2 -14

2. Iain S. Stewart and Ted, (2013) Earth stories: context and narrative in the communication of popular geoscience, Proceedings of the Geologist' Association 124 (2013), pp. 699 – 714

3. Shingo Nagamatsu, Toshinari Nagasaka, Yuichiro Usuda and Saburo Ikeda, (2009) How can the "Coping Capacity of the Local Community Against Disasters" be Evaluated ?, 74th Research Report 2009, National Research Insittute for Earth Science and Disaster Prevention, Japan

[Figure]

Please also note the supplement to this comment:
http://www.nat-hazards-earth-syst-sci-discuss.net/nhess-2016-15/nhess-2016-15-AC2-supplement.pdf

───────────────────────────────
[Figure]

**QUESTION BEFORE LANDSLIDES EDUCATION**

Location        : (school name)
Instruction    : Please tick (√) in every box

A.   **Demographics**
1. Age          :
2. Sex          :  ☐ Male   ☐ Female

B.   **Knowledge on Landslides**
3. Did landslides happen near to your house?  ☐ Yes   ☐ No
4. Landslides happened mostly because of
   ☐ Heavy rain
   ☐ Earth quake
5. Which will trigger landslides
   ☐ Strong wind
   ☐ Heavy rain
   ☐ Lightning
6. Among the below natural disaster, which often in your place?
   ☐ Flood
   ☐ Landslides
   ☐ Thunder
   ☐ Typhoon
7. Is it important to know about mechanism of landslide?   ☐ Yes   ☐ No
8. Should I know often getting more knowledge on landslides  ☐ Yes   ☐ No
9. Landslides cause losses of money and casualties           ☐ Yes   ☐ No

**QUESTION ON LANDSLIDES STUDY**

10. Fill in the blank with the number

1. Thermometer is for measuring air temperature
2. Rain gauge is for measuring rainfall intensity
3. Anemoscope is for pointing the wind direction
4. Compass is for getting the direction
5. Anemometer is for measuring wind speed

**Fig. 1.** Question Before Landslides Education(English)

**Fig. 2.** Question After Landslides Education(English)

[Figure]

SOALAN SOAL SELIDIK KAJIAN TANAH RUNTUH

10. Isikan kotak kosong tersebut dengan nombor di bawah.

1. Termometer atau jangka suhu merupakan alat untuk mengukur suhu udara
2. Alat pengukur curah hujan digunakan untuk mengukur air hujan.
3. Alat penentuan arah angin digunakan untuk petunjuk arah angin
4. Kompas adalah alat petunjuk arah
5. Alat pengukur kecepatan angin digunakan untuk mengukur kecepatan angin.

SOALAN SOAL SEBELUM SELIDIK KAJIAN TANAH RUNTUH

Lokasi :
Arahan : Sila tandakan (√) pada kotak yang disediakan

A. Latar Belakang Demografi
1. Umur :
2. Jantina : ☐ Lelaki ☐ Perempuan

B. Pengetahuan Tentang Tanah Runtuh
3. Pernahkah anda mengalami kejadian tanah runtuh di tempat anda? ☐ Ya ☐ Tidak
4. Kejadian tanah runtuh di Malaysia biasanya disebabkan oleh:
☐ Hujan lebat
☐ Gempa bumi
5. Tanah runtuh terjadi kerana:
☐ Serangan Semut
☐ Pengaliran Air
☐ Tiupan Angin
6. Antara bencana alam berikut, manakah yang sering terjadi di tempat anda?
☐ Banjir
☐ Tanah runtuh
☐ Kilat
☐ Angin dan taufan
7. Penting untuk saya mengetahui kejadian tanah runtuh. ☐ Ya ☐ Tidak
8. Penting untuk saya sentiasa menambah ilmu pengetahuan
mengenai tanah runtuh. ☐ Ya ☐ Tidak
9. Kejadian tanah runtuh melibatkan nyawa dan kerugian. ☐ Ya ☐ Tidak

**Fig. 3.** Question Before Landslides Education(Malay)

SOALAN SOAL **SELEPAS** SELIDIK KAJIAN TANAH RUNTUH

Lokasi :
Arahan : Sila tandakan (/) pada kotak yang disediakan

**A. Latar Belakang Demografi**
1. Umur :
2. Jantina : ☐ Lelaki ☐ Perempuan

**B. Pengetahuan Tentang Tanah Runtuh**
3. Pernahkah anda mengalami kejadian tanah runtuh di tempat anda? ☐ Ya ☐ Tidak
4. Kejadian tanah rumah di Malaysia biasanya disebabkan oleh:
☐ Hujan lebat
☐ Gempa bumi
5. Tanah rumah terjadi kerana:
☐ Serangan Semut
☐ Pengaliran Air
☐ Tiupan Angin
6. Antara bencana alam berikut, manakah yang sering terjadi di tempat anda?
☐ Banjir
☐ Tanah rumah
☐ Kilat
☐ Angin dan taufan
7. Penting untuk saya mengetahui kejadian tanah rumah. ☐ Ya ☐ Tidak
8. Penting untuk saya sentiasa menambah ilmu pengetahuan mengenai tanah rumah. ☐ Ya ☐ Tidak
9. Kejadian tanah rumah melibatkan nyawa dan kerugian. ☐ Ya ☐ Tidak

**C. Sikap**
10. Saya lebih bersedia untuk menghadapi kejadian tanah rumah pada masa akan datang. ☐ Ya ☐ Tidak
11. Saya mengambil tahu tentang kejadian tanah rumah kerana ia penting untuk menyelamatkan nyawa. ☐ Ya ☐ Tidak

SOALAN SOAL SELIDIK KAJIAN TANAH RUNTUH
12. Isikan kotak kosong tersebut dengan nombor di bawah.

1. Termometer atau jangka suhu merupakan alat untuk mengukur suhu udara
2. Alat pengukur curah hujan digunakan untuk mengukur air hujan.
3. Alat pecerminan arah angin digunakan untuk penunjuk arah angin
4. Kompas adalah alat penunjuk arah
5. Alat pengukur kecepatan angin digunakan untuk mengukur kecepatan angin.
13. Sekiranya diberi pilihan, adalah anda berminat untuk mengikuti kelas seperti ini, pada masa akan datang?
1. Ya ☐
2. Tidak ☐

**Fig. 4.** Question After Landslides Education(Malay)

**Fig. 5** Early warning system to collect data from monitoring sites where the locations are N05° 32.918' E101° 20.749' and N05° 36.042' E101° 35.546'.

**Fig. 5.** Early Warning System

[Figure]

Fig. 6 If the curve crosses over the cautious critical or critical line, the light in the lamp will turn from green colour to yellow colour or red colour respectively. Moreover, the buzzer will sound if the light turns to red.

**Fig. 6.** Early Warning System User Interface

[Figure]

Fig. 7 User can run the simulation by inputting soil properties, slope angle and rainfall intensity to understand the behaviour of water table and the stability of slope after rainfall.

[Figure]

**Fig. 7.** Slope Stability Prediction(Simulator)

[Figure]

Fig. 8 Hazard Map Management System provides landslides historical information in Malaysia for users to understand when and where landslides occurred in the past for future cautious.

**Fig. 8.** Hazard Map Management System

---

## Author Comment (AC3) · 19 Feb 2016

Please refer to the Fig. 1 for literacy rate(aged 25 years and older) in South East Asia Countries(ASEAN) as the reference for our reply "Furthermore, in rural areas, some parents are not well-educated.".

[Figure]

Education Attainment of the Population aged 25 years and older (%) in South East Asia Countries(ASEAN)

| Country Name | Primary | Lower Secondary | Upper Secondary | Tertiary Education |
|---|---|---|---|---|
| | | 2009 | | |
| Brunei | 98 | 75(secondary) | 75(secondary) | 11 |
| Cambodia | 20.1 | 9.2 | 4.2 | - |
| Indonesia | 30.58 | 14.45 | 20.34 | 7.5 |
| Laos | 68.4 | 75 | - | 3 |
| Malaysia | 23.7 | 17.5 | 31.2 | 18.3 |
| Myanmar | 83.94 | 39.57(secondary) | 39.57(secondary) | - |
| Philippines | 89.4 | 59.9(secondary) | 59.9(secondary) | 28.1 |
| Singapore | 96.8 | 95.2 | 74.6 | 63.6 |
| Thailand | 89.7 | 72.7(secondary) | 72.7(secondary) | 45.8 |
| Vietnam | 98.5 | 84.96(secondary) | 84.96(secondary) | 51.73 |

\* Source: ASEAN State of Education Report 2013

\*- means not available

\*\*secondary means the survey does not separate lower or upper secondary

**Fig. 1.** Education Attainment of the Population aged 25 years and older in South East Asia Countries(ASEAN)